# COVID-19-Related Neuropathic Pain: A Systematic Review and Meta-Analysis

**DOI:** 10.3390/jcm12041672

**Published:** 2023-02-20

**Authors:** Laura Dawn Williams, Panagiotis Zis

**Affiliations:** 1Medical School, University of Sheffield, Sheffield S10 2RX, UK; 2Medical School, University of Cyprus, Nicosia 2029, Cyprus; 3Second Department of Neurology, School of Medicine, Attikon University Hospital, National and Kapodistrian University of Athens, 124 62 Athens, Greece

**Keywords:** neuropathic pain, COVID-19, long COVID

## Abstract

Introduction: SARS-CoV-2, responsible for the coronavirus disease (COVID-19) pandemic, may impact other systems apart from the respiratory system, including the nervous system. In this systematic review, we aimed to establish the prevalence and determinants of neuropathic pain amongst COVID-19-infected individuals. Methodology: A literature search in the PubMed database was performed and 11 papers were eligible for inclusion in this systematic review and meta-analysis. Results: The pooled prevalence of COVID-19-related neuropathic pain was 6.7% (95% CI: 4.7–9.5%) for hospitalised patients during the acute phase and 34.3% (95% CI: 14.3–62%) for long COVID patients. The identified risk factors for COVID-19-related neuropathic pain development included depression, COVID-19 severity and azithromycin use. Conclusions: Neuropathic pain is a very common symptom in long COVID, indicating the urgency for further research in this direction.

## 1. Introduction

SARS-CoV-2, which was responsible for the coronavirus disease (COVID-19) pandemic, can lead to severe lung damage and has high mortality rates [1]. Apart from the respiratory system, a SARS-CoV-2 infection may impact multiple systems, including the cardiovascular [2] and the nervous systems. Long COVID syndrome is a term used to describe resisting symptoms months after the initial infection. A variety of symptoms have been described as part of long COVID syndrome. One under-researched complication is the development of neuropathic pain following a COVID-19 infection [3]. 

Neuropathic pain is caused by somatosensory nervous system diseases or lesions [4]. Many risk factors for the pathogenesis of neuropathic pain have been identified, including systemic diseases, autoimmune diseases, incisions, channelopathies, metabolic disorders, viral infections and chemotherapy drug-induced mechanisms [5,6,7,8]. Although the exact molecular mechanisms are still to be discovered [9], neuropathic pain is estimated to affect up to 10% of the general population [10].

Neuropathic pain has the potential to become severely debilitating, hugely impacting the quality of life of an individual [4]. Associated autonomic symptoms can also inflict long-term disabilities [11]. Mobility issues can limit socialisation and independence, leading to depression, anxiety and sleep disorders [12]. Additionally, neuropathic pain often presents with analgesic/opioid medication resistance [13], leaving treatment options limited and tailored to the individual. Subsequently, this highlights the necessity for neuropathic pain assessments across different cohorts, including COVID-19.

The exact prevalence of COVID-19-related neuropathic pain is unestablished to date [14]. COVID-19 is able to interact with the nervous system via multiple mechanisms, promoting nociceptor excitability and, therefore, neuropathic pain [15]. Furthermore, the evidence suggests that neuropathic pain is occasionally an immediate COVID-19 complication; in other cases, it is more delayed [3].

In this systematic review, we aimed to establish the prevalence and determinants of COVID-19-related neuropathic pain by systematically applying eligibility criteria and performing a quantitative analysis.

## 2. Methodology

### 2.1. Protocol and Registration

This review was registered in PROSPERO, an international database of prospectively registered systematic reviews in health and social care. The registration number for this review is CRD42022330381.

### 2.2. Literature Search Strategy

A systematic literature search in the PubMed database was performed on 16 May 2022 using two medical subject heading terms that had to be present in the title or the abstract. Term A was “Allodynia OR hyperalgesia OR hypoalgesia OR itchiness OR burning OR neuropathic OR ‘painful cold’ OR ‘electric shock’ OR tingling OR ‘pins and needles’ OR itchiness OR itching OR hypoesthesia” and term B was “COVID or SARS-CoV-2 or Coronavirus”. Human subjects, the English language and full-text filters were applied. The reference lists of eligible papers and relevant reviews were also meticulously searched to include further studies reporting on neuropathic pain related to COVID-19.

Decisions about the search term use involved the PICO (population, intervention, comparison and outcome) framework [16], highlighted in Table 1. The PICO framework aids the formulation/discovery of answers to healthcare research questions in an evidence-based manner [17]. In this review, how to best detect COVID-19-related neuropathic pain articles was deciphered from the PICO framework. This helped to answer the research aims regarding the prevalence.

### 2.3. Inclusion/Exclusion Criteria

Articles eligible to be included in this review had to meet the following criteria: (i) papers describing patients with neuropathic pain as defined by the International Association for the Study of Pain [18] that were clearly linked to COVID-19; (ii) human adult subjects involved; and (iii) the article was written in English.

The exclusion criteria for this review comprised: (i) a description of neuropathic pain that was not within the primary aims of the study; (ii) case series/cohorts with fewer than 10 patients; (iii) non-original articles (reviews, medical hypotheses, letters to the editor, etc.); and (iv) articles describing non-neuropathic pain.

In descending order, title screening, abstract screening, full-text screening and reference screening were implemented to filter out non-relevant papers. This left the relevant articles suitable for inclusion. Decisions regarding which papers were eligible for inclusion were made by two researchers. Reference screening was implemented using Google Scholar, purely to identify any papers that may have been missed during the original search. However, reference screening yielded no additional literature.

### 2.4. Data Extraction

Following the identification of eligible articles, information covering the phenotypes, incidence, natural history, treatments and narrative summary for COVID-19-related neuropathic pain was collated. Colour-coding assigning the content of each paper was utilised to achieve a narrative summary. Other factors such as the COVID-19 severity, patient demographics and comorbidities were extracted and taken into consideration. Neuropathy types and the reported time between an initial COVID-19 infection and neuropathic pain manifestation were extracted. A meta-analysis was conducted, in which the article details regarding long COVID were included. A full-text scrutiny of each included article aided the risk factor identification, contributing to the overall field of COVID-19-related neuropathic pain.

### 2.5. Synthesis of Results

The prevalence, descriptives, frequencies and other determinants were investigated via the extracted data. The means, 95% confidence intervals and meta-analysis (*p* < 0.05) were calculated, in line with the Preferred Reporting Items for Systematic Reviews and Meta-analyses (PRISMA) guidelines.

### 2.6. Statistical Analysis

Aggregated data were used where possible in this review. Statistically pooled proportion calculations were conducted in R language using the default settings of the “meta” package and the “metaprop” function with a random fixed effects model [19]. The heterogeneity was evaluated using the I^2^ statistic and corresponding forest plot [20]. The I^2^ statistic suggested whether the study heterogeneity or chance were responsible for the variation. Negative I^2^ values were set as equal to 0 and the values ranged between 0% and 100% [20]. The heterogeneity could be quantified as low, moderate and high, with upper limits of 25%, 50% and 75% for I^2^, respectively [20]. Occasionally, a meta-analysis was not suitable for the data. Resultantly, a narrative approach was taken.

### 2.7. Assessment of Bias

Bias assessments adapted from [21] assessed the methodological quality of the included articles via a checklist. The response rate, sample representativeness, data collection method, measures used, sampling techniques and statistical methods were covered by nine questions. Each question was labelled as “0” or “1” (0 = low risk of bias; 1 = high risk of bias). The total scores ranged from 0–9 and were categorised as follows [22]: 7–9: high risk of bias; 4–6: moderate risk of bias; 0–3: low risk of bias.

Appendix A details this bias assessment.

## 3. Results

### 3.1. COVID-19-Related Neuropathic Pain

Following the systematic literature search strategy described above, 384 papers were initially identified. However, 275 were excluded during the abstract screening and 98 were excluded during the full-text eligibility screening. Therefore, 11 articles met the eligibility and were finally included in this review. The screening and inclusion process is highlighted in Figure 1 (PRISMA chart). The characteristics of the included papers are summarised in Appendix A.

### 3.2. Epidemiology

Figure 2 shows the total pooled prevalence of COVID-19-related neuropathic pain in COVID-19 sufferers following the meta-analysis of 8 eligible studies [23,24,25,26,27,28,29,30], which assessed a total of 1059 patients. The pooled prevalence of neuropathic pain was 23.2% (95% CI: 11.0–42.4%). As also shown in the respective funnel plot (Figure 2), the heterogeneity was high amongst these studies (I^2^ = 94%).

As two of the studies provided data for the acute phase and six studies for the long COVID phase, a subgroup analysis was performed.

Figure 3 shows the total pooled prevalence of COVID-19-related neuropathic pain in hospitalised patients following the meta-analysis of 2 eligible studies [24,25], which assessed a total of 438 patients. The pooled prevalence during hospitalisation (acute phase of COVID-19) was 6.7% (95% CI: 4.7–9.5%). As also shown in the respective funnel plot (Figure 3), there was a high heterogeneity across these studies (I^2^ = 0.0%).

Figure 4 shows the total pooled prevalence of COVID-19-related neuropathic pain in long COVID patients following the meta-analysis of 6 eligible studies [23,26,27,28,29,30], which assessed a total of 621 patients. The pooled prevalence of COVID-19-related neuropathic pain in long COVID patients was 34.3% (95% CI: 14.3–62%). As also shown in the respective funnel plot (Figure 4), the heterogeneity was high amongst these studies (I^2^ = 94%).

### 3.3. Risk Factors for Neuropathic Pain

#### 3.3.1. Demographics

Neither age nor gender were associated with a neuropathic pain symptom presence overall [23,24,25,26,27,28,29,30,31,32]. However, the male gender was suggested to be a predictor of increased pain severity (odds ratio 4.3; 95% CI: 1.8–10.6) [24] and a younger age was proposed to contribute towards new-onset neuropathic pain following a severe COVID-19 infection. In their observational study, Ojeda et al. showed that patients with new-onset pain had a mean age of 60 years versus a mean age of 68 years for patients not developing new-onset pain (*p* = 0.005; univariate analysis) [28].

#### 3.3.2. COVID-19 Severity

A neuropathic pain symptom presence correlated with the COVID-19 severity [24] for hospitalised COVID-19 patients. Moreover, those with an increased COVID-19 severity who had to be hospitalised in an intensive care unit were more likely to acquire peripheral nerve injuries due to a prone positioning for an extended amount of time [25].

Although the results on whether the COVID-19 severity was linked to neuropathic pain in long COVID were contradictory, the majority of the studies reported that severe COVID-19 increased the likelihood of COVID-19 neuropathic pain development up to 6.3 times [28,29,31,32,33].

#### 3.3.3. Treatment with Azithromycin

Several studies suggested that a treatment with azithromycin increased the likelihood of developing neuropathic pain following a COVID-19 infection by up to 5.4 times [28,31]. Azithromycin may induce neuronal cell autophagy [31]. Concerningly, the odds of COVID-19-related neuropathic pain significantly increased following an azithromycin treatment [28].

#### 3.3.4. Depression, Anxiety and Psychological Distress

A two-way relationship between pain and the mental state exists. Depression, anxiety and psychological distress may occur in those with neuropathic pain following a COVID-19 infection [29]; depression was also considered to be an independent predictor of COVID-19-related neuropathic pain in [31] as it may increase the risk of pain by 4.5 times.

### 3.4. Locations

Different areas of the body have been reportedly affected by neuropathic pain after a COVID-19 infection. The locations reported were the leg/calf, knees, hands, muscles, neck, shoulders, lower abdomen, head and feet [24,27]. Neuropathic pain presented diffusely across the whole body in a few cases [24]. Other cases displayed COVID-19-related neuropathic pain within the extremities [31,32]. An interesting neuropathic pain syndrome that was described was trichodynia, which occurred within 4 weeks of the initial COVID-19 infection [33].

### 3.5. Natural History

The time between a COVID-19 initial infection and neuropathic pain development ranged from 1 month to 15 months for long COVID patients [23,28]. Upon discharge, half of the long COVID patients went on to report new-onset pain 4 weeks later [28]. Conclusions are difficult, as the papers did not investigate similar time lengths [23,26,29]. However, all these papers showed that COVID-19-related neuropathic pain could be acute (4 weeks) or delayed (4 months+). Therefore, COVID-19-related neuropathic pain should still be considered to be plausible in delayed circumstances.

## 4. Discussion

This review aimed to establish the prevalence of neuropathic pain following a COVID-19 infection. Although neuropathic pain can be a symptom in the acute phase of COVID-19 (the pooled prevalence for hospitalised patients was 6.7%), it is more common in the context of long COVID syndrome as it affects more than 1 in 3 sufferers (the pooled prevalence was 34.3%). The current literature suggests that long COVID syndrome affects one in three of those infected with COVID-19 [34]. Therefore, one in nine of those infected with COVID-19 will develop neuropathic pain. This is a significant number of people, considering the millions who have been infected with COVID-19 (Mercer and Salit, 2021).

The COVID-19-related neuropathic pain predictors that proved to be significant in the literature included the COVID-19 severity (hospitalised cohorts) [24], prolonged ICU prone positioning [25], azithromycin use and depression [31]. Therefore, psychological interventions to treat depression might prove useful in reducing the risk of COVID-19-related neuropathic pain. This is quite significant, considering 61% of COVID-19 patients were seeking neuropathic pain treatments at the follow-up [23].

Our findings should be interpreted with caution, given a few limitations. First, a gender bias impacted certain samples [24,25], possibly preventing detectable differences from being uncovered. Future replications require equally balanced samples. Second, most studies lacked control groups. Third, our search was restricted to a single database (PubMed). Despite the fact that PubMed is the largest medical database where the majority of (if not all) high-quality studies appear, there was a small risk that we might have missed a few relevant papers. Finally, publication bias may occur in systematic reviews and could have undermined the validity of the results. This was reflected in the significant heterogeneity among the studies of long COVID patients.

In conclusion, in this review we highlighted the prevalence and predictors of COVID-19-related neuropathic pain. More research is needed in the future to explore these findings in more detail, considering that several patients have been left unable to function due to neuropathic pain following COVID-19 [27].

## Figures and Tables

**Figure 1 jcm-12-01672-f001:**
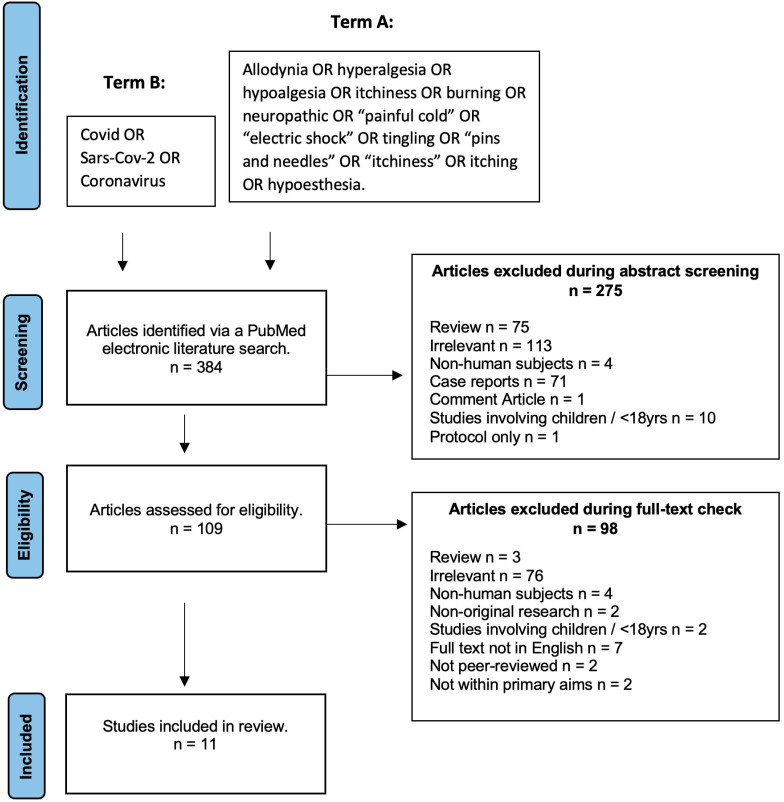
PRISMA chart displaying the screening and inclusion process for the COVID-19-related neuropathic pain systematic review and meta-analysis.

**Figure 2 jcm-12-01672-f002:**
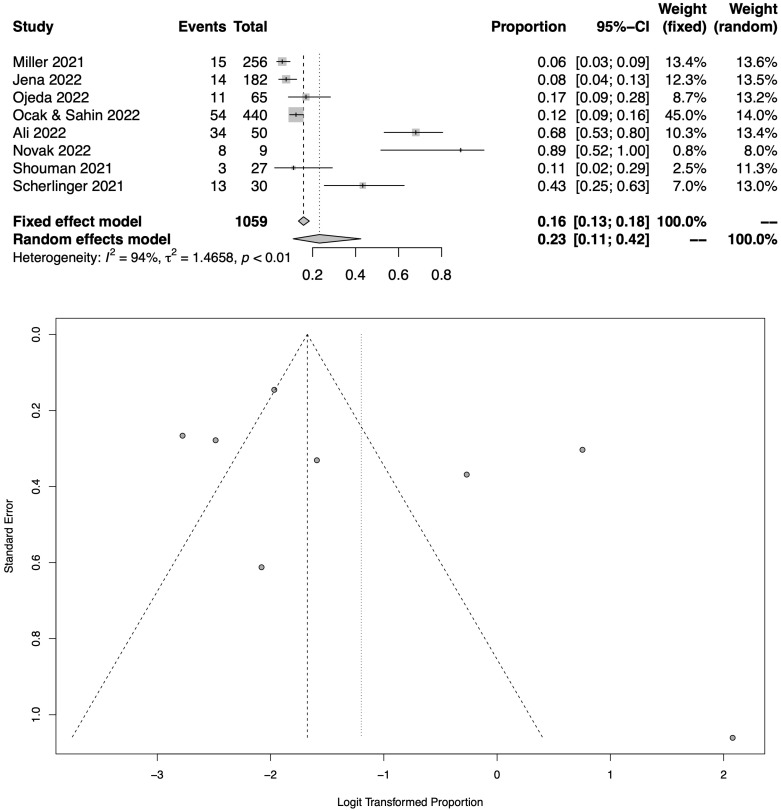
Meta-analysis results (all RCTs with available data) as illustrated in the forest plot and the respective funnel plot regarding the pooled prevalence estimation for COVID-19-related neuropathic pain amongst COVID-19 sufferers [23,24,25,26,27,28,29,30].

**Figure 3 jcm-12-01672-f003:**
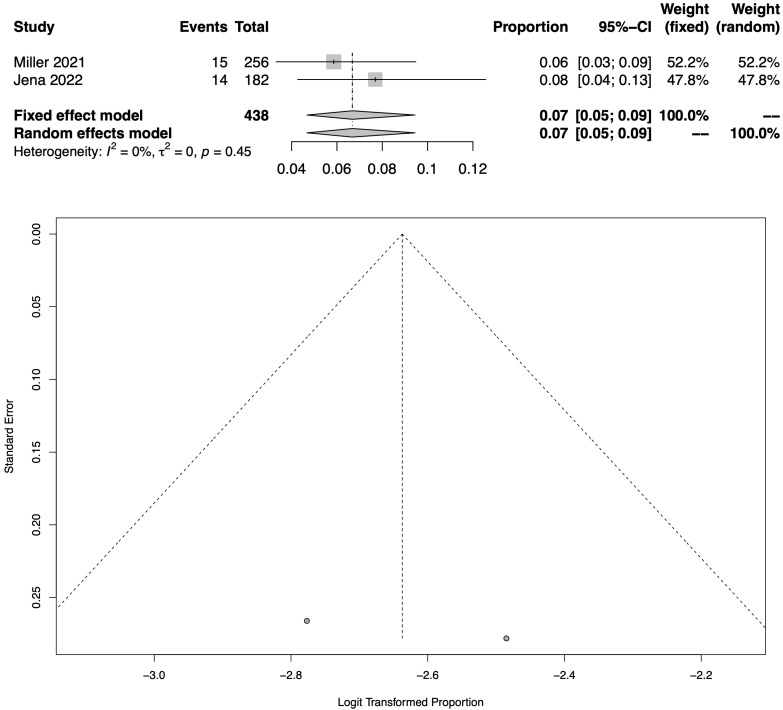
Meta-analysis results (all RCTs with available data) as illustrated in the forest plot and the respective funnel plot regarding the pooled prevalence estimation for COVID-19-related neuropathic pain amongst hospitalised patients [24,25].

**Figure 4 jcm-12-01672-f004:**
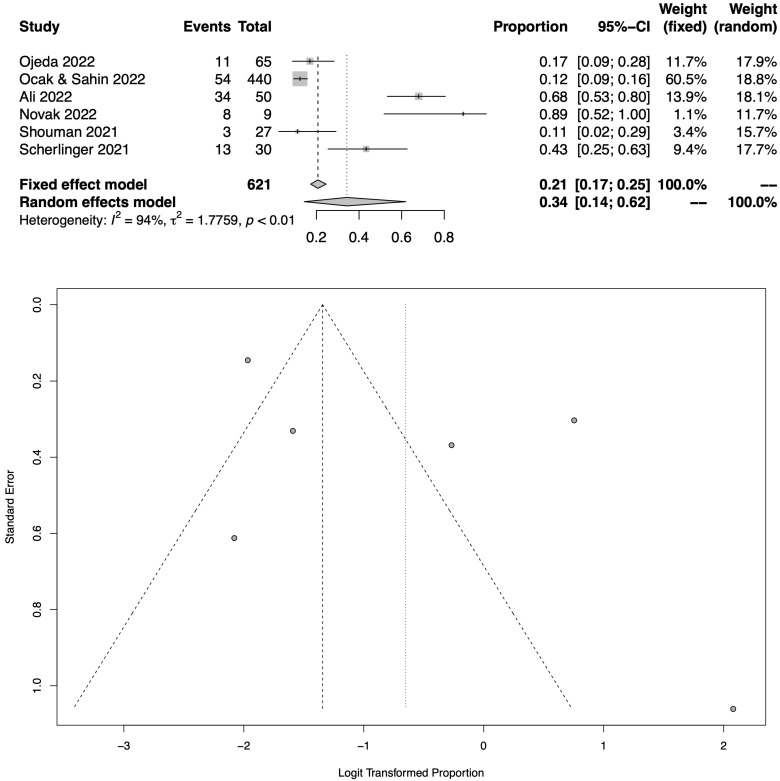
Meta-analysis results (all RCTs with available data) as illustrated in the forest plot and the respective funnel plot regarding the pooled prevalence estimation for COVID-19-related neuropathic pain amongst long COVID cohorts [23,26,27,28,29,30].

**Table 1 jcm-12-01672-t001:** PICO criteria applied to literature/systematic search strategy.

Population	Adults Only (≥18 Years of Age)
Intervention	Neuropathic pain following COVID-19 infection.
Comparison	Characteristics; Time (between COVID-19 infection and neuropathic pain manifestation); Gender dominance; Types of symptoms described; Long COVID syndrome.
Outcome	Prevalence (COVID-19-related neuropathic pain).

## Data Availability

The data presented in this study are available on request from the corresponding author.

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
