# Peer review of "COVID-19-Related Neuropathic Pain: A Systematic Review and Meta-Analysis"

_jcm, 2023, doi:10.3390/jcm12041672_

Round 1

Reviewer 1 Report

Estimated Editors of JCM,

Estimated Authors,

I've read with great interest the present paper entitled "COVID-19 Related Neuropathic Pain: A Systematic Review and Meta-analysis". The present paper focuses on a very interesting topic, and the summary of available evidence may represent a considerable contribution to the daily management of COVID-19 by international medical professionals. As a consequence, the present paper may represent not only an interesting but also useful piece of information.

This manuscript has the following problems need to be solved:

1. It is unclear whether the present SRMA has been performed or not following the PRISMA Guidelines (in case, which iteration?); some piece of information would be, in this case, lacking, most notably PICO, and Summary Table with summary of included studies; moreover, the qualitative assessment of included studies is not reported even though Authors claim that it is included in supplementary material (not made available to the present reviewer)

2. Authors have included only Forrest plots of the included studies, and their design could be improved by including a cumulative + subgroup analysis (i.e. instead of two estimates of prevalence, please perform a single analysis with calculation for a cumulative estimate, and two subgroup analysis with estimates for long Covid (now figure 3) and hospitalized patients (now figure 1). 

3. Reporting of I2 estimates is unclear. I2 is estimated to be very low (0%) in the sole subgroup of hospitalized individuals (2 studies!); in the subgroup long COVID, th I2 is in turn very high (94%), and this subgroup includes the overall majority of all included patients. Therefore, double check the statement from from row 142 onwards. 

4. Please provide funnel and radial plots for assessment of publication and small study bias

5. In the present study, you opted in for implementing a "random fixed effects model", that may be quite appropriate (at least for the present reviewer) but you're reporting both fixed and random models: please, make a choice and stick with your decision.

6. Figure 1 has to be reworked and cleaned from various marks

7. Risk factors (section 3.3) are not quantitatively reported by any table, and should be more precisely reported, at least as a series of  supplementary tables.

Please fix the aforementioned issues before sharing a revised version of this contribution.

Author Response

Dear Reviewer, we are uploading our point by point responses to your comments

Reviewer 2 Report

The article entitled “COVID-19 Related Neuropathic Pain: A Systematic Review and Meta-analysis” aims to establish the prevalence and determinants of neuropathic pain amongst COVID-19 infected individuals. The authors state that the prevalence of COVID-19 related neuropathic pain was 6.7% (95 CI 4.7% - 9.5%), for hospitalized patients during the acute phase and 34.3% (95 CI 14.3% - 62%) for Long-COVID patients. The identified risk factors for COVID-19 related neuropathic pain development included depression, COVID-19 19 severity and azithromycin use.

Overall, it is a topic of interest given the high global prevalence of covid-19 patients. The manuscript is clear, relevant for the field and presented in a well-structured manner. Likewise, the conclusions are consistent with the evidence and arguments presented.

However, there are a few limitations and comments that need to be pointed out:

-       An important limitation is that the systematic review is restricted to a single database (Pubmed) and consequently, the results of the search may be insufficient.

-       The specific definition of neuropathic pain used by the authors to select the papers should be stated, since in the search equation, words such hypoesthesia (which may not be considered as neuropathic pain) were included.

-       It would be of great interest to add an explicative table refering all the included articles and summarizing the patient´s characteristics, number of included patients, neuropathic pain assessment/definition, time since covid-19 to development of pain, prevalence of the condition and neuropathic pain related clinical /risk factors.

-       The authors state that a bias assessments adapted from Hoy et al (2012) would be performed. However, no information about risk-of-bias analysis is given in the results section. Even if the details are given in the supplementary material, a summary of the overall bias assessment should be added to the manuscript (result section).

-       To conclude, there are some interesting facts that, if available, could be mentioned in the results or discussion sections: Is there any common and/or differential pattern between the neuropathic pain developed during on-hospital stay and that developed later (long-covid patients)? Was any of the risk factors more frequent in any of these two cohorts or related to a specific topography?

Author Response

(The authors gave the same response as above.)

Round 2

Reviewer 1 Report

Estimated Authors,

I've appreciated your efforts to improve the overall quality of this study.

Please be aware that some shortcomings still remains: for instance, the study has been performed through the analysis of a single database (i.e. Pubmed). Moreover, I'm still noticing some issues with the Figure 1.

I'm attaching hereby Figure 1 as I'm assessing it. Please understand my concerns regarding the eventual reproduction in the final version.

However, as most of the issues have been solved, I'm endorsing the eventual acceptance, leaving the fixing of Figure 1 in the editorial stage.
